# Learning from Algorithm Feedback: One-Shot Sat Solver Guidance with GNNs

**Jan Tönshoff**[*]
RWTH Aachen University

**Martin Grohe**
RWTH Aachen University

## Abstract

Boolean Satisfiability (SAT) solvers are foundational to computer science, yet their performance typically hinges on hand-crafted heuristics. This work introduces Reinforcement Learning from Algorithm Feedback (RLAF) as a paradigm for learning to guide SAT solver branching heuristics with Graph Neural Networks (GNNs). Central to our approach is a novel and generic mechanism for injecting inferred variable weights and polarities into the branching heuristics of existing SAT solvers. In a single forward pass, a GNN assigns these parameters to all variables. Casting this one-shot guidance as a reinforcement learning problem lets us train the GNN with off-the-shelf policy-gradient methods, such as GRPO, directly using the solver's computational cost as the sole reward signal. Extensive evaluations demonstrate that RLAF-trained policies significantly reduce the mean solve times of different base solvers across diverse SAT problem distributions, achieving more than a 2x speedup in some cases, while generalizing effectively to larger and harder problems after training. Notably, these policies consistently outperform expert-supervised approaches based on learning handcrafted weighting heuristics, offering a promising path towards data-driven heuristic design in combinatorial optimization.

## 1 Introduction

Solving computationally hard combinatorial problems, such as Boolean satisfiability (Sat), remains a cornerstone of computer science and is critical to diverse domains such as verification, planning, and cryptography (Biere et al., 2021). Complete search algorithms are of particular importance, as they are guaranteed to find a solution if one exists or prove unsatisfiability otherwise. The runtime of these classical algorithms heavily depends on hand-crafted heuristics to navigate the solution space, for example, by determining variable assignments during the search. Such heuristics are often rigid and hard to adapt to specific instance distributions without extensive expert knowledge and tuning. Machine learning offers a compelling alternative: Augmenting the heuristic components of classical search algorithms with trainable functions allows us to construct adaptable solvers. Specifically, reinforcement learning (RL) can train these extended solvers to learn improved, distribution-specific heuristics in a data-driven manner without direct expert supervision.

In this work, we study how to leverage RL-trained Graph Neural Network (GNNs) to improve branching heuristics of Sat solvers. Our main contributions are as follows: First, we introduce a novel and generic method for integrating variable-wise weights into the branching heuristics of existing Sat solvers. Secondly, we construct a GNN-based policy that assigns a weight and polarity to each variable in one forward pass. This one-shot setting enables a single GNN pass to influence every branching decision, avoiding costly repeat passes. Thirdly, we phrase the task of inferring weights and polarities that reduce the solver's cost as an RL problem. The reward signal is directly obtained from the observed computational cost of the guided Sat solver, requiring no expert supervision. We refer to this training paradigm as **R**einforcement **L**earning from **A**lgorithm **F**eedback (RLAF). Finally, we demonstrate empirically that modern RL techniques, such as GRPO (Shao et al., 2024), are capable of training effective RLAF policies for different base solvers. The learned policies substantially reduce solver runtimes, generalize to harder problems after training, outperform supervised baselines, and appear to capture solver-agnostic structural properties of Sat problems.

---

[*]Correspondence to `toenshoff@informatik.rwth-aachen.de`

**Algorithm 1** DPLL Solver

```
1: Input: Formula φ
2: function SOLVE(φ)
3:     # Simplify formula
4:     φ ← UNIT-PROPAGATION(φ)
5:     φ ← PURE-LITERAL-ELIMINATION(φ)
6:
7:     if φ = ∅: return SAT
8:     if ∅ ∈ φ: return UNSAT
9:
10:    # Decide next branching variable
11:    ℓ ← PICK-LITERAL(φ)
12:    return SOLVE(φ ∧ {ℓ}) ∨ SOLVE(φ ∧ {¬ℓ})
13: end function
```

**Algorithm 2** Decision Heuristic

```
1: Input: Formula φ
2: function PICK-LITERAL(φ)
3:     x̂ ← argmaxₓ SCORE(x)
4:     return x̂ if PICK-SIGN(x̂) else ¬x̂
5: end function
```

**Algorithm 3** Guided Decision Heuristic

```
1: Input: Formula φ, Parameters 𝒲 = (w, p)
2: function PICK-LITERAL-GUIDED(φ, 𝒲)
3:     x̂ ← argmaxₓ w(x) · SCORE(x)
4:     return x̂ if p(x̂) = 1 else ¬x̂
5: end function
```

Figure 1: DPLL SAT solver and branching heuristics. Algorithm 1: A DPLL SAT solver performs backtracking search to solve a given CNF formula $\phi$. At each search step, the formula is simplified through unit propagation and pure literal elimination before selecting the next branching literal. Algorithm 2: Branching heuristics are often implemented by choosing the variable that maximizes some hand-crafted scoring function. Algorithm 3: We propose to extend existing branching heuristics by incorporating given variable weights into the branching decisions that scale the associated score of each variable. We additionally choose the sign of each literal according to a provided polarity.

## 1.1 BACKGROUND

**SAT Solving**   A Boolean formula in Conjunctive Normal Form (CNF) is a conjunction of clauses $\phi = C_1 \wedge \cdots \wedge C_m$, each clause being a disjunction of one or more literals $C_j = (\ell_{j,1} \vee \cdots \vee \ell_{j,k})$. We denote by $\mathrm{Var}(\phi) = \{x_1, \ldots, x_n\}$ the set of Boolean variables of $\phi$. The Boolean SAT problem is to decide whether or not there exists a satisfying assignment $\alpha : \mathrm{Var}(\phi) \rightarrow \{0, 1\}$ that satisfies all clauses of a given formula $\phi$. This problem is well-known to be NP-complete and naturally arises in a wide range of applications (Biere et al., 2021). Modern SAT solvers predominantly stem from the Davis-Putnam-Logemann-Loveland (DPLL) algorithm, a backtracking search approach enhanced by unit propagation and pure literal elimination. Algorithm 1 provides a pseudocode description of a DPLL SAT solver. Many extensions of this general idea have been proposed to scale SAT solvers to larger, industrial instances. In particular, Conflict-Driven Clause Learning (CDCL) solvers significantly extend the DPLL framework by introducing clause learning and non-chronological backtracking. A common property of DPLL-derived solvers is the importance of the branching heuristic that picks the next branching literal in each search step (line 11 in Algorithm 1). Various branching heuristics have been proposed, and which heuristic performs best often depends on the structure of the given SAT formula $\phi$ (Kullmann, 2021). Customizing branching heuristics towards a specific distribution of inputs generally requires expert knowledge and significant trial and error.

**Reinforcement Learning**   Reinforcement learning (RL) aims to learn policies for sequential decision-making problems where an agent interacts with an environment to learn through trial and error. An RL problem is usually formalized as a Markov Decision Process (MDP), which is defined as a tuple $(\mathcal{S}, \mathcal{A}, P, R)$, where $\mathcal{S}$ is the set of states, $\mathcal{A}$ is the set of possible actions, $P$ denotes the transition probabilities between states and $R$ is the reward function. In probabilistic settings, policies $\pi$ are stochastic mappings from states to distributions over actions, i.e., $\pi(a|s)$ indicates the probability of selecting action $a$ in state $s$. More generally, for continuous action spaces, i.e. $\mathcal{A} = \mathbb{R}$, $\pi(a|s)$ is the probability *density* that a policy assigns to an action $a$. The primary objective in RL is to determine an optimal policy $\pi^*$ that maximizes the expected cumulative discounted reward $\mathbb{E}_\pi \left[ \sum_{t=0}^{\infty} \gamma^t R(s_t, a_t) \right]$, where $\gamma \in [0, 1)$ represents a discount factor that emphasizes earlier rewards. This formulation is the basis for various RL algorithms. Recently, RL algorithms based on policy gradients, such as PPO (Schulman et al., 2017) and GRPO (Shao et al., 2024) have been used extensively to fine-tune LLMs from human feedback (RLHF, Christiano et al. (2017); Ouyang et al. (2022)) and from verifiable rewards (RLVR, Lambert et al. (2024); Guo et al. (2025)).

## 1.2 RELATED WORK

Leveraging deep learning in the context of combinatorial optimization (CO) problems has emerged as a major area of research (Cappart et al., 2021) and has been applied to a wide range of problems such as combinatorial graph problems (Khalil et al., 2017), SAT solving (Selsam et al., 2019), Mixed-Integer Programming (Khalil et al., 2022), and Constraint Satisfaction Problems (Tönshoff et al., 2023). Here, we primarily focus on work that aims to enhance SAT solvers with (graph) neural networks. One line of work suggests using predictions of predefined variable properties to guide SAT solver branching heuristics. Selsam & Bjørner (2019) train a GNN to predict whether variables belong to an UNSAT core. The branching heuristic is then guided by periodically resetting the solver's VSIDS scores to the GNN's predictions, thus making the guidance specific to VSIDS-based CDCL solvers and dependent on careful tuning of the reset frequency. Wang et al. (2024) predict whether literals occur in the backbone of satisfiable formulas and use these predictions to set the polarity of variables. Another line of work explores purely RL-based training for enhancing branching heuristics, eliminating the need for expert supervision. Kurin et al. (2020) uses Q-learning to train GNNs end-to-end as branching policies to minimize solver runtime, and Cameron et al. (2024) propose Monte Carlo Forest Search for guiding early branching decisions in SAT Solvers on UNSAT problems. Both methods require one GNN forward pass per guided branching decision, which creates a significant bottleneck as the GNN usually requires orders of magnitude more runtime than classical branching heuristics. As a consequence, both methods only leverage the learned guidance in a few early solver decisions. Further related work is proposed by Han (2020), who accelerate cube-and-conquer solvers with supervised learning, Liu et al. (2024), who suggest improving clause deletion heuristics in CDCL SAT solvers with GNNs, and Zhai & Ge (2025), who use RL to speed up parallelized divide-and-conquer solvers.

## 2 RLAF-GUIDED SAT SOLVERS

### 2.1 GUIDED BRANCHING HEURISTICS

We modify existing SAT solvers to incorporate external variable weights into their branching heuristic. Let some base SAT solver be given. We assume that this solver is a DPLL-derived backtracking search algorithm. We further assume that the branching heuristic is implemented by first selecting a variable $\hat{x} = \text{argmax}_x \texttt{Score}(x)$ that maximizes some variable scoring function $\texttt{Score}$ before picking a literal sign according to some secondary heuristic, as illustrated in Algorithm 2. Many existing branching heuristics, such as VSIDS and look-ahead heuristics, fit the generic algorithm pattern while relying on different definitions of variable scores. Note that these scores usually depend on the current partial assignment of the search as well as information extracted in previous search steps, such as encountered conflicts. We can modify this decision heuristic to incorporate additional variable weights $w : \text{Var}(\phi) \to \mathbb{R}_{>0}$ for the given input formula $\phi$:

$$\hat{x} = \text{argmax}_x w(x) \cdot \texttt{Score}(x) \tag{1}$$

These weights are passed to the modified solver as additional input and modulate its branching heuristic by scaling the variable-wise scores. In this manner, we can inject prior knowledge of variable importance into the solver's branching decisions without sacrificing its original heuristic. Naturally, choosing a useful variable weighting $w$ by hand is difficult. Instead, our focus is on learning to infer effective variable weights from the input formula's structure using a deep neural network.

In addition to these weights, we may also specify a mapping $p : \text{Var}(\phi) \to \{0, 1\}$ that assigns a polarity $p(x)$ to each variable $x$. When $x$ is chosen as a decision variable, the polarity determines which value is assigned to $x$ first. Specifying polarities for variables is a common function for modern SAT solvers, and well-chosen values can have a significant impact on run time, especially on satisfiable instances. In this work, we will infer variable-wise polarities alongside the variable weights $w$ with a learned GNN model. Overall, the modified solver $\texttt{Solve}(\phi, \mathcal{W})$ takes as input a CNF formula $\phi$ as well as a variable parameterization $\mathcal{W} = (w, p)$ that assigns a weight $w(x) \in \mathbb{R}_{>0}$ and polarity $p(x) \in \{0, 1\}$ to each variable $x \in \text{Var}(\phi)$.

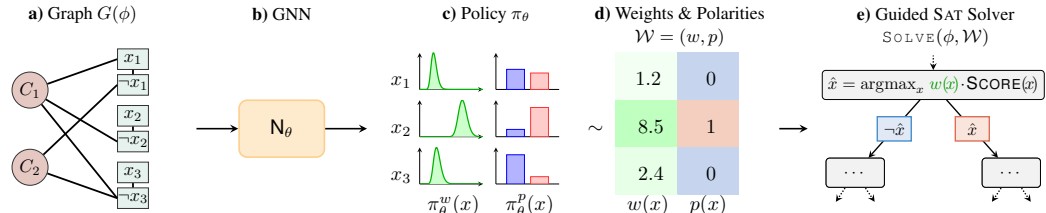

Figure 2: **a)** The input formula $\phi$ is modeled as a graph $G(\phi)$. **b)** The graph is processed by a trainable GNN and outputs a parameterization policy $\pi_\theta(\phi)$. **c)** The policy $\pi_\theta(\phi)$ consists of independent variable-wise weight (LogNormal) and polarity (Bernoulli) distributions. **d)** A variable parameterization $\mathcal{W} = (w, p)$ is sampled from $\pi_\theta(\phi)$, mapping each variable $x$ in $\phi$ to a weight $w(x) \in \mathbb{R}_{>0}$ and polarity $p(x) \in \{0, 1\}$. **e)** A guided SAT solver incorporates the parameterization $\mathcal{W}$ to guide its branching heuristic.

## 2.2 Graph Representation and Architecture

Our goal is to map an instance $\phi$ to advantageous variable weights and polarities with a neural network. A natural approach is to map $\phi$ to a suitable graph representation $G(\phi) = (V(\phi), E(\phi))$ that captures the instance's structure. This graph can then be processed by a GNN that extracts structural information in a trainable manner. We represent $\phi$ as a standard "Literal-Clause Graph" proposed in prior work Selsam et al. (2019). Note that this choice is modular; other graph representations have also been suggested in the literature and could also be used. We process this graph with a trainable GNN model $N_\theta$ that performs message passing to extract latent structural embeddings for every vertex. Here, $\theta$ represents the vector that contains all trainable model parameters. The output of $N_\theta$ is a mapping $y : \text{Var}(\phi) \to \mathbb{R}^2$ that assigns two real numbers to each variable in the input formula $\phi$. The full model details are provided in Section A.

## 2.3 Guidance Policy

For a given input formula $\phi$, we map the output of the GNN $N_\theta$ to a policy $\pi_\theta(\phi)$ from which a variable parameterization $\mathcal{W} \sim \pi_\theta(\phi)$ can be sampled. Recall that for a given SAT instance the GNN $N_\theta$ outputs a mapping $y : \text{Var}(\phi) \to \mathbb{R}^2$ that associates every variable $x \in \text{Var}(\phi)$ with two real numbers $\mu(x), \rho(x) \in \mathbb{R}$, $[\mu(x), \rho(x)] = y(x)$. These outputs are used to parameterize variable-wise weight and polarity distributions, respectively. Concretely, for each variable $x$ in $\phi$ we define its weight policy $\pi_\theta^w(x)$ as a Log-Normal distribution over positive real weights:

$$\pi_\theta^w(x) = \text{LogNormal}(\mu(x), \sigma^w) \tag{2}$$

Here, the inferred parameter $\mu(x) \in \mathbb{R}$ is used as the log-mean of the distribution, and $\sigma^w \in \mathbb{R}_{>0}$ is a hyperparameter. We employ the Log-Normal distribution because it naturally models unimodal distributions over strictly positive real numbers, ensuring that the sampled weights remain valid multiplicative scaling factors without requiring artificial clipping or constraints. While we observed reasonable training convergence with alternative parameterizations, such as Truncated Normal and Poisson distributions, the Log-Normal formulation proved simpler and more robust.

Analogously, we define a variable's polarity policy $\pi_\theta^p(x)$ as a Bernoulli distribution where the probability is obtained by applying a sigmoid function to $\rho(x)$:

$$\pi_\theta^p(x) = \text{Bernoulli}(\text{Sigmoid}(\rho(x))). \tag{3}$$

The complete variable parameterization policy $\pi_\theta$ is then defined as the joint distribution of $\pi_\theta^w(x)$ and $\pi_\theta^p(x)$ over all variables:

$$\pi_\theta(\phi) = \pi_\theta^w(x_1) \times \pi_\theta^p(x_1) \times \cdots \times \pi_\theta^w(x_n) \times \pi_\theta^p(x_n). \tag{4}$$

We sample a variable parameterization $\mathcal{W} = (w, p) \sim \pi_\theta$ from this distribution in one shot by independently sampling a weight $w(x) \sim \pi_\theta^w(x)$ and polarity $p(x) \sim \pi_\theta^p(x)$ for each variable $x$ in parallel. The probability density $\pi_\theta(\mathcal{W}|\phi)$ of $\mathcal{W}$ can then be factorized as

$$\pi_\theta(\mathcal{W}|\phi) = \prod_x \pi_\theta^w(w(x)|\phi) \cdot \pi_\theta^p(p(x)|\phi). \tag{5}$$

**a)** Sample Parameterizations | **b)** Run Solver | **c)** Cost & Advantage | **d)** Optimize $\mathcal{L}_{\text{PPO}}$

Cost $\quad \hat{A}$

$$\phi \longrightarrow \pi_\theta(\phi) \begin{array}{l} \widetilde{\sim} \mathcal{W}_1 \\ \sim \mathcal{W}_2 \\ \sim \mathcal{W}_3 \end{array} \longrightarrow \begin{array}{l} \texttt{Solve}(\phi, \mathcal{W}_1) \\ \texttt{Solve}(\phi, \mathcal{W}_2) \\ \texttt{Solve}(\phi, \mathcal{W}_3) \end{array} \longrightarrow \begin{array}{ll} 150 & -1.39 \\ 128 & 0.91 \\ 132 & 0.49 \end{array} \longrightarrow \begin{array}{l} \pi_\theta(\mathcal{W}_1 \,|\, \phi) \downarrow \\ \pi_\theta(\mathcal{W}_2 \,|\, \phi) \uparrow \\ \pi_\theta(\mathcal{W}_3 \,|\, \phi) \uparrow \end{array}$$

Figure 3: Learning to accelerate a SAT solver with GRPO: **a)** For a given training formula $\phi$ sample multiple variable parameterizations i.i.d. from the current policy $\pi_\theta(\phi)$. **b)** Run the SAT solver on $\phi$ with each parameterization. **c)** Map the cost of each solver run (i.e. the number of decisions) to the normalized group-relative advantage $\hat{A}(\phi, \mathcal{W})$. **d)** Optimize the model weights $\theta$ to maximize $\mathcal{L}_{\text{PPO}}$ to shift the policy towards faster parameterizations.

Note that all trainable weights of the GNN model have a partial derivative with respect to $\pi_\theta(\mathcal{W}|\phi)$ for a given $\mathcal{W}$, which enables us to train with policy-gradient methods such as GRPO. During training, we sample multiple $\mathcal{W}$ i.i.d. from $\pi_\theta(\phi)$ and use the variance of the observed solver runtimes to compute our training signal, as explained in Section 2.4. At test time, we do not sample randomly from $\pi_\theta(\phi)$ but simply use the mode $\hat{\mathcal{W}}$, which deterministically chooses the most probable weight and polarity for each variable $x$. This eliminates a source of variance when testing and, on average, yields better results than sampling at random from the learned policy.

## 2.4 POLICY OPTIMIZATION

Our aim is to learn a policy GNN that guides the SAT solver towards lower computational costs on a given distribution of SAT instances. Formally, let $\Omega$ be some training distribution of SAT problems. The objective is to learn model weights $\theta$ that minimize the expected solver cost when applying the learned policy to instances sampled from $\Omega$:

$$\theta^* = \arg\min_\theta \mathbb{E}_{\phi \sim \Omega, \mathcal{W} \sim \pi_\theta(\phi)} \left[ \texttt{Cost}(\phi, \mathcal{W}) \right]. \tag{6}$$

Here, $\texttt{Cost}(\phi, \mathcal{W})$ is defined as the number of decisions required when running $\texttt{Solve}(\phi, \mathcal{W})$, which is the primary target metric we aim to minimize. We can view this objective as an RL problem by modeling the process of choosing $\mathcal{W}$ as a single-step Markov Decision Process (MDP) where the input formula $\phi$ is viewed as the state, and a single-step episode unfolds by choosing a variable parameterization $\mathcal{W}$ as the action. Once the action is taken, the environment transitions immediately to a terminal state, yielding a reward $R(\phi, \mathcal{W}) = -\texttt{Cost}(\phi, \mathcal{W})$ that is the negative of the solver's cost (e.g., number of decisions). Note that we also experimented with directly using CPU time as a cost measure, but found this to yield less stable training due to the performance variance caused by noisy CPU utilization.

We leverage Group Relative Policy Optimization (GRPO) (Shao et al., 2024) to learn a policy for this RL problem. GRPO is a simplification of Proximal Policy Optimization (PPO) (Schulman et al., 2017) that eliminates the need for learning an additional value network. The initial model weights $\theta_0$ are sampled at random. GRPO updates these model weights in iterations $k \in \{1, \ldots, K\}$. In iteration $k$, we first sample a batch of training instances from $\mathcal{F} = \{\phi_1, \ldots, \phi_N\} \sim \Omega^N$ from the given training distribution. For each such formula $\phi_i$ we sample $M$ variable parameterizations $\mathcal{W}_{i,1}, \ldots, \mathcal{W}_{i,M} \sim \pi_{\theta_{k-1}}(\phi_i)$ i.i.d. from the current policy. We then run $\texttt{Solve}(\phi_i, \mathcal{W}_{i,j})$ for all $i, j \in [N] \times [M]$ and measure the corresponding cost and reward. The group-relative advantage is then defined as

$$\hat{A}_{i,j} = \frac{R(\phi_i, \mathcal{W}_{i,j}) - \text{mean}(\mathbf{R}_i)}{\text{std}(\mathbf{R}_i)} \tag{7}$$

where $\mathbf{R}_i = \{R(\phi_i, \mathcal{W}_{i,j}) \mid j \in \{1, \ldots, M\}\}$ is the set of all rewards collected for the same instance $\phi_i$. The main objective is to maximize the clipped policy update function for each training instance $\phi_i$:

$$\mathcal{L}_{\text{PPO}}(\theta \mid \phi_i) = \frac{1}{M} \sum_j \left[ \min \left( r_{i,j}(\theta)\hat{A}_{i,j}, \text{clip}(r_{i,j}(\theta), 1-\epsilon, 1+\epsilon)\hat{A}_{i,j} \right) \right]. \tag{8}$$

Here, $\epsilon \in (0, 1)$ is a hyperparameter, and $r_{i,j}(\theta)$ is defined as the probability ratio of the new policy and the policy learned in the previous GRPO iteration:

$$r_{i,j}(\theta) = \frac{\pi_\theta(\mathcal{W}_{i,j}|\phi_i)}{\pi_{\theta_{k-1}}(\mathcal{W}_{i,j}|\phi_i)} \tag{9}$$

This objective aims to adjust the policy such that actions (e.g., variable parameterizations) with high advantage become more likely while avoiding excessively large distribution shifts by clipping the objective at a probability ratio determined by $\epsilon$. The full training objective combines $\mathcal{L}_{\text{PPO}}$ with an additional term that penalizes the KL divergence relative to the previous model weights $\theta_{k-1}$ to stabilize training further:

$$\mathcal{L}(\theta \mid \phi_i) = \mathcal{L}_{\text{PPO}}(\theta \mid \phi_i) - \beta \cdot \text{KL}\left(\pi_\theta(\phi_i), \pi_{\theta_{k-1}}(\phi_i)\right). \tag{10}$$

Here, the weight $\beta \geq 0$ is an additional hyperparameter. Starting from the previous model weights $\theta_{k-1}$, we learn updated model weights $\theta_k$ by performing stochastic gradient ascent for a fixed number of steps to maximize this objective function for all training instances. This overall process repeats in the next round of GRPO. In the appendix, Algorithm 4 provides a complete formal specification of our training.

## 2.5 TRAINING SETUP

We are utilizing GRPO as an online RL algorithm to learn the parameters of our policy GNN directly from observed solver costs. As a consequence, we train with the SAT solver in-the-loop and make $M \cdot N$ calls to the solver per GRPO iteration. With our default parameters ($N = 100$, $M = 40$) we make 4000 SAT solver calls in each iteration. This imposes the practical constraint to train on a distribution $\Omega$ of SAT problems where this number of solver calls is possible in an acceptable time on the underlying hardware. The work presented here intends to be a small-scale demonstration of RLAF as a training paradigm, and all training is performed on machines with one (multi-core) CPU and one GPU. Therefore, the training data in our experiments is chosen so that each instance is solvable by the given base solvers within a fraction of a second. In future work, the hardness and size of the training problems can be scaled up substantially by leveraging a distributed compute cluster for collecting the SAT solver feedback. Crucially, we demonstrate in Section 3 that after training, the learned policies do generalize to significantly harder and larger problems. The reliance on comparatively easy training problems is therefore not a significant limitation for learning effective GNN-guidance with RLAF.

## 3 EXPERIMENTS

In our experiments, we aim to answer two primary research questions: (i) Can RLAF train GNN-based guidance policies that shorten solver runtimes and generalize to harder formulas? (ii) How do RLAF-trained policies fare against guidance based on learning predefined notions of variable importance in a supervised manner? Furthermore, we want to understand whether the learned policies capture known variable properties after training and whether the policies learned with different solvers are related or solver-specific.

**Solvers** We conduct experiments with two distinct base solvers: The well-known CDCL solver Glucose (Audemard & Simon, 2017) and the DPLL solver March (Heule et al., 2005). Glucose uses the EVSIDS branching heuristic and is comparatively strong on structured problems, while March uses a look-ahead branching heuristic and is among the best-known solvers for random instances. Full details on the extended solvers are provided in Section A.3. As our focus is a prototype-scale demonstration of RLAF as a training paradigm, we leave integration with state-of-the-art CDCL solvers for future work aimed at large-scale industrial settings.

**Data** We consider three well-known classes of SAT problems with significantly different structures to study how well RLAF can adapt the base solvers to each of them. In Section B.1 we provide full details on the data generation and dataset statistics. **Random 3SAT**: We define $3\text{SAT}(n)$ as the distribution of uniformly random 3SAT instances with $n$ variables and clause-to-variable ratio of $4.26$, which is approximately the critical density where the instances transition from SAT to UNSAT. The training data consists of 20K instances sampled from $3\text{SAT}(200)$. We test on larger instances with

Table 1: Results on test instances. All metrics are averaged across the respective test sets. The mean number of decisions is rounded to the nearest whole number. For results with RLAF, we include the time required for the GNN forward pass in the total runtime. We highlight numbers in bold when they are the best value achieved for the respective base solver.

| Data | | | Glucose | | Glucose + RLAF | | March | | March + RLAF | |
| Distribution | Result | Count | Decisions | Time (s) | Decisions | Time (s) | Decisions | Time (s) | Decisions | Time (s) |
|---|---|---|---|---|---|---|---|---|---|---|
| 3SAT(300) | SAT | 103 | 341,418 | 6.67 | **121,184** | **1.85** | 2,893 | 0.25 | **2,389** | **0.23** |
| | UNSAT | 97 | 725,812 | 15.49 | **508,676** | **8.21** | 11,783 | **1.01** | **11,757** | 1.04 |
| 3SAT(350) | SAT | 108 | 1,568,289 | 48.76 | **805,035** | **18.88** | 16,546 | 1.64 | **11,702** | **1.19** |
| | UNSAT | 92 | 3,628,268 | 132.28 | **3,136,552** | **82.84** | 52,287 | **5.14** | **51,846** | 5.16 |
| 3SAT(400) | SAT | 89 | 9,638,668 | 598.35 | **4,447,304** | **186.70** | 64,296 | 7.27 | **47,992** | **5.51** |
| | UNSAT | 111 | 22,130,692 | 1,895.62 | **20,808,043** | **1,112.71** | 245,064 | **27.49** | **242,499** | 27.51 |
| 3COL(400) | SAT | 77 | 15,519 | 0.36 | **6,988** | **0.22** | 926 | 0.22 | **598** | **0.21** |
| | UNSAT | 123 | 70,692 | 1.99 | **34,920** | **0.81** | 10,563 | 2.61 | **5,954** | **1.57** |
| 3COL(500) | SAT | 91 | 82,758 | 2.61 | **35,901** | **1.05** | 7,689 | 2.55 | **4,754** | **1.68** |
| | UNSAT | 108 | 460,881 | 17.47 | **363,278** | **12.24** | 100,811 | 33.34 | **60,321** | **20.52** |
| 3COL(600) | SAT | 87 | 606,598 | 25.03 | **339,378** | **11.59** | 63,512 | 27.16 | **42,862** | **18.71** |
| | UNSAT | 113 | 3,092,344 | 193.96 | **2,811,133** | **155.23** | 754,720 | 313.57 | **461,639** | **197.12** |
| CRYPTO(20) | UNSAT | 100 | 51,294 | 1.16 | **3,541** | **0.15** | 1,203 | 0.82 | **390** | **0.41** |
| CRYPTO(15) | UNSAT | 100 | 225,447 | 5.74 | **64,150** | **1.40** | 52,282 | 34.56 | **8,257** | **6.40** |
| CRYPTO(10) | UNSAT | 100 | 3,753,850 | 162.45 | **1,520,075** | **64.95** | 679,864 | 467.38 | **230,905** | **169.22** |

$n \in \{300, 350, 400\}$, where we sample 200 instances for each size $n$. **Graph Coloring**: We consider the distribution 3COL($n$) of SAT problems that decide 3-colorability for random Erdős-Rényi graphs with $n$ vertices. We set the edge probability such that the expected vertex degree is $4.67$, which is approximately the critical density for 3-colorability where hard instances are common (Zdeborová & Krząkała, 2007). We train on 20K instances sampled from 3COL(300) on 200 larger problems each for $n \in \{400, 500, 600\}$. **Cryptographic**: We further include highly structured instances arising from SAT-based decryption attacks (Soos et al., 2009). We define CRYPTO($n$) as the distribution of SAT instances generated for decrypting the HiTag2 cipher (Courtois et al., 2009) with $n$ given help bits. Note that these instances are harder for smaller values of $n$ and are mostly UNSAT. We train on 20K instances from CRYPTO(22) and test on harder problems with $n \in \{20, 15, 10\}$.

**Hyperparameters** We train a different model for each solver and each SAT problem class. We configure the GNN with 10 layers with embedding dimension $d = 256$. We train for $K = 2000$ GRPO iterations, each consisting of 50 training steps of SGD. In every iteration, we use $N = 100$ training formulas and collect feedback for $M = 40$ variable parameterizations for each formula.

## 3.1 MAIN RESULTS

Table 1 provides the main results for both Glucose and March on our test sets. We observe that GNN-guided training with RLAF consistently accelerates the given base solver. The margin of improvement depends on the base solver and the class of problem instances. For 3SAT(400) problems, RLAF-guidance reduces the mean runtime of Glucose by 69% and 41% for satisfiable and unsatisfiable instances, respectively. Similar improvements are observed for satisfiable 3-coloring problems as well as cryptographic instances. For unsatisfiable coloring instances with 600 vertices, the runtime of Glucose is reduced by around 24%. The smallest margin of improvement is observed for the March solver on unsatisfiable 3SAT instances. While March+RLAF does need fewer decisions on average to solve this class of problems, the runtime is worse, as the small improvement does not compensate for the additional runtime overhead of the GNN forward pass. It is known that lookahead DPLL solvers like March are very strong baselines for unsatisfiable random instances, so this result is not surprising. For more structured problem classes, RLAF is able to accelerate the March solver substantially on both satisfiable and unsatisfiable instances. We emphasize that the wall-clock runtime of the GNN forward pass is included in the runtime measurements with RLAF-guidance. For the instances used here, this runtime is generally around 0.1 seconds or less, which is negligible compared to the solver runtimes on harder problems. We refer to Section B.4 in the appendix for extended results that report the GNN overhead in detail and plot the cumulative number of solved instances over time. Overall, these results demonstrate that RLAF is able to train GNN-based solver guidance and that relying on comparatively easy problems for efficient training does not prevent the learned policy from generalizing to more complex problems at test time.

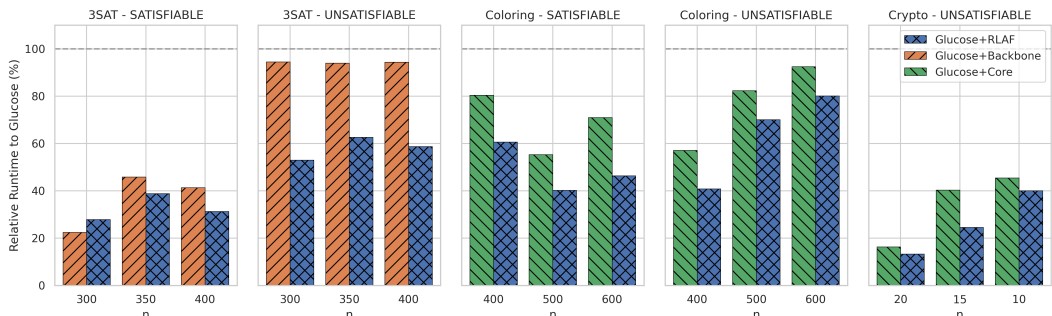

Figure 4: Runtimes relative to the base solver Glucose for RLAF and supervised approaches based on Backbones and UNSAT cores. **Less is better**.

## 3.2 COMPARISON TO SUPERVISED APPROACHES

Prior work suggests predicting predefined variable properties, such as UNSAT core Selsam & Bjørner (2019) or backbone Wang et al. (2024) membership, in a supervised manner, and then transforming model predictions into variable weights and polarities for solver guidance. Here, we aim to compare how guidance learned with RLAF compares to this approach. Note that the notions of UNSAT cores and backbones are only sensible training targets for some instance distributions. Backbones can only be non-empty on satisfiable instances, and even for satisfiable graph coloring problems, all backbones are empty due to the permutation symmetry of the vertex colors. Furthermore, on our UNSAT 3SAT training instances, we observed that the UNSAT core extracted by SAT solvers contained all variables on almost all instances, yielding a training target that is effectively constant. Due to these limitations, we use the 3SAT instances to evaluate the effectiveness of predicting the backbone, while we use the graph coloring and cryptographic instances to compare RLAF to core-based solver guidance. For a fair comparison, we use the same GNN architecture used to train with RLAF and train a separate model for each instance distribution. The transformation that maps the backbone/core predictions to variable weights is tuned separately for each instance distribution on the corresponding validation set. Full details about the setup of this comparison are provided in Section B.5. We also provide an additional comparison of RLAF to Graph-Q-SAT Kurin et al. (2020), an unsupervised baseline using Q-Learning, in Section B.6.

Figure 4 compares the results for Glucose in terms of the relative wall-clock runtime compared to the base solver. Overall, the policy learned with RLAF significantly outperforms solver guidance based on both UNSAT core and backbone predictions by achieving a smaller relative runtime. The backbone-based heuristic outperforms RLAF only on satisfiable 3SAT instances with 300 variables, but not on larger problems. On unsatisfiable 3SAT problems, the backbone-guided heuristic performs substantially worse. RLAF also outperforms core-based guidance for both graph coloring and cryptographic SAT problems. Overall, these results demonstrate that pure RL-based learning with RLAF can yield more effective solver guidance than predicting handcrafted notions of variable importance in a supervised manner.

## 3.3 EXPLORING LEARNED VARIABLE WEIGHTS

We further aim to gain insights into the weight distributions learned through RLAF. In particular, we investigate whether the policies learned with different base solvers are related and whether they capture predefined variable properties, such as backbone and UNSAT core membership. To this end, Figure 5 compares the weights for 5000 randomly selected variables from the corresponding validation sets. Specifically, we plot the expected variable weight $\mathbb{E}[w(x)]$ for the Glucose-trained policy on the x-axis and plot the corresponding value for the March-trained policy on the y-axis. We also report the Pearson correlation coefficient ($r$) for these weights to quantify their correlation. For 3SAT, we only plot variables from satisfiable instances and additionally indicate whether each variable belongs to its instance's backbone. Likewise, we focus on unsatisfiable instances for 3COL and CRYPTO and indicate if a variable occurs in the UNSAT core extracted for the experiment in Section 3.2.

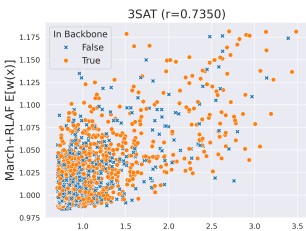 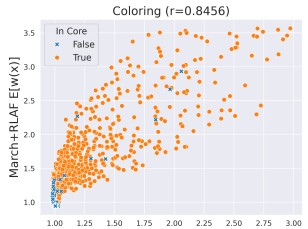 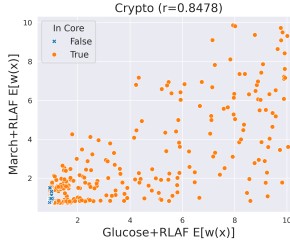

Figure 5: Weight correlation between policies learned with different solvers. For each instance distribution, we randomly sample $5,000$ variables $x$ from the corresponding validation set and plot the expected variable weight $E[w(x)]$ for the policies learned with either base solver. The color further indicates the backbone or UNSAT core membership of each variable.

We observe that the variable weights of the two policies are generally correlated, with a Pearson correlation coefficient $r$ between $0.73$ and $0.85$. This indicates that the learned weightings capture structural properties that are inherent to the variables and accelerate the search across different solvers. We further observe that for the 3COL and CRYPTO instances, the variables with high weights are predominantly members of the UNSAT core. For these problem instances, the RLAF-based training therefore self-discovered weight policies that correlate to existing handcrafted heuristics while performing better, as demonstrated in Section 3.2. For the 3SAT instances, we do not observe a clear correlation between the learned weight policies and backbone membership, showing that in this case, the trained models express functions that, while effective, do not resemble this particular handcrafted heuristic.

## 4 DISCUSSION

We introduced RLAF as a paradigm for training GNN-based policies that guide the branching heuristics of SAT solvers. Our work contributes (i) a generic mechanism for injecting variable weights into branching heuristics, (ii) a formulation of weight selection as a one-shot RL problem, (iii) a way to leverage GNNs as trainable policies in this setting, and (iv) experimental evidence that GRPO can learn policies that reduce the computational cost of different base solvers. In our empirical studies, the learned policies generalize to larger and harder instances, and consistently surpass supervised baselines that rely on handcrafted variable properties. Moreover, policies trained with different base solvers converge toward similar structural signals, suggesting that RLAF is capturing inherent information about SAT instance structure that is not specific to the underlying base solver.

The current implementation is designed as a small-scale prototype optimized for training on relatively simple formulas using moderate hardware. This constraint allows for only comparatively simple problems to be solved quickly enough to collect sufficient solver feedback on local CPU cores in each GRPO iteration. Expanding the system to leverage distributed computing resources would enable the incorporation of larger and harder SAT problems during training, potentially leading to better guidance policies. GNN scalability also remains a bottleneck, particularly during training, as processing large industrial instances imposes significant memory and computational demands. At the same time, the expressive power of our network, a standard message-passing GNN, is bounded by color refinement (Morris et al., 2019; Xu et al., 2019). Combining higher expressivity and scalability remains a critical challenge for applying GNNs to large, structured problem instances.

Finally, the proposed methodology is not strictly limited to SAT solvers. Branching heuristics are critical components not only in SAT solving but also for Mixed-Integer Programming (MIPs) and Constraint Satisfaction Problems (CSPs). More broadly, implementing any kind of selection heuristic as the $\mathrm{argmax}$ of some scoring function is a generic pattern of algorithm design found across many domains. For any such algorithm, one can introduce external multiplicative weights that guide the heuristic and then phrase the task of inferring effective weights as an RL problem. In this work, we have demonstrated that this general methodology can be leveraged in the context of SAT solving. Translating it to other domains and algorithms remains as future work.

REPRODUCIBILITY STATEMENT

All experimental code is included in the supplementary material (`https://github.com/toenshoff/RLAF`), including scripts for data generation and preprocessing. We also provide detailed information on the hyperparameters required to reproduce our results in Section B in the appendix.

ACKNOWLEDGMENTS

The project was funded by the European Union (ERC, SymSim, 101054974). Views and opinions expressed are however those of the author(s) only and do not necessarily reflect those of the European Union or the European Research Council. Neither the European Union nor the granting authority can be held responsible for them.

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

## A   METHOD DETAILS

### A.1   GRAPH REPRESENTATION AND ARCHITECTURE

We represent a formula $\phi$ as a graph $G(\phi) = (V(\phi), E(\phi))$. This is a standard "Literal-Clause Graph" used in prior work, such as NeuroSAT Selsam et al. (2019). Formally, the vertices of this graph $V(\phi) = \text{Lit}(\phi) \cup \text{Cls}(\phi)$ are the literals and clauses of $\phi$. The edges $E(\phi) = E_{LC}(\phi) \cup E_{LL}(\phi)$ connect literals with the clauses they occur in and with the opposing literal of the same variable:

$$E_{LL}(\phi) = \{(x, \neg x) \mid x \in \mathcal{X}(\phi)\} \tag{11}$$

$$E_{LC}(\phi) = \bigcup_{C \in \text{Cls}(\phi)} \{(C, \ell) \mid \ell \in C\} \tag{12}$$

### A.2   MESSAGE-PASSING NEURAL NETWORK

For each vertex $v \in \text{Lit}(\phi) \cup \text{Cls}(\phi)$ we obtain an initial embedding $h^0(v) \in \mathbb{R}^d$:

$$h^0(v) = \mathbf{Enc}(\log(\deg(v) + 1)). \tag{13}$$

Here $d$ is the latent embedding dimension of the model, and $\mathbf{Enc}$ is a trainable 2-layer MLP that is applied to the log-normalized degree of $v$.

The GNN model then stacks $L \in \mathbb{N}$ message passing layers. For $t \in \{1, \ldots, L\}$, the $t$-th layer takes as input the previous embedding $h^{t-1}$ and outputs a refined embedding $h^t$ by performing a message pass. This message pass is split into two phases. First, each clause $c \in \text{Cls}$ aggregates information from its associated literals:

$$h^{t+1}(c) = \mathbf{U}_{\text{Cls}}\left(h^t(c), \bigoplus_{\ell \in c} h^t(\ell)\right). \tag{14}$$

Here, $\mathbf{U}_{\text{Cls}}$ is a trainable MLP, and $\bigoplus$ is an order-invariant aggregation. Throughout all experiments, we use element-wise mean for aggregation. In the second phase, each literal $\ell \in \text{Lit}$ aggregates the updated embeddings from the clauses it occurs in:

$$h^{t+1}(\ell) = \mathbf{U}_{\text{Lit}}\left(h^t(\ell), h^t(\neg \ell), \bigoplus_{c, \ell \in c} h^{t+1}(c)\right). \tag{15}$$

Here, $\mathbf{U}_{\text{Lit}}$ is another trainable MLP that additionally also takes the embedding of the opposing literal $\neg \ell$ as input. This model architecture is conceptually similar to that of NeuroSAT. One major difference is that we use a more standard fixed-depth feed-forward GNN instead of a recurrent model. Note that all MLPs used in our model have two layers, and the hidden layer is always SiLU-activated and has hidden dimension $2d$. The final output is a variable embedding $y : \text{Var}(\phi) \to \mathbb{R}^2$, which is obtained by concatenating the two literal embeddings associated with each variable $x$ and then applying a final 2-layer MLP $\mathbf{Dec}$:

$$y(x) = \mathbf{Dec}([h^L(x), h^L(\neg x)]). \tag{16}$$

Note that we choose $\mathbf{Dec}$ as a 2-layer MLP with input dimension $2d$, hidden dimension $2d$, and output dimension 2. No activation is applied to the output, and the weights and biases of the final layer of $\mathbf{Dec}$ are initialized as zeros. This ensures that at the beginning of training, the initial GNN $N_{\theta_0}$ assigns $\mu(x) = 0$ and $\rho(x) = 0$ to all variables. We found this to be a stable configuration for initializing training. In particular, $\mu(x) = 0$ ensures that the log-normally distributed weight policy $\pi_{\theta_0}^w(x)$ has a mode of approximately 1 for all variables while $\rho(x) = 0$ ensures that the polarity of each variable is initially distributed uniformly.

### A.3   SAT SOLVER DETAILS

#### GLUCOSE

Glucose (Audemard & Simon, 2009) is a popular CDCL solver based on Minisat (Eén & Sörensson, 2003). Our modification is based on Glucose 4.2.1 (Audemard & Simon, 2017)[1]. Like many other

---

[1] https://github.com/audemard/glucose

CDCL solvers, Glucose uses the Variable State Independent Decaying Sum (VSIDS) heuristic for branching. Each variable $x$ is assigned an activity score $\text{activity}(x)$ that reflects its involvement in conflicts. When a conflict occurs, the activity scores of variables involved are increased by a constant $\Delta$, i.e.,

$$\text{activity}(x) \leftarrow \text{activity}(x) + \Delta. \tag{17}$$

Periodically, all activity scores are multiplied by a decay factor $\beta$ (where $0 < \beta < 1$):

$$\text{activity}(x) \leftarrow \beta \cdot \text{activity}(x). \tag{18}$$

The activity then effectively serves as the SCORE function from Algorithm 2. Note that in practice, CDCL solvers commonly use *exponential* VSIDS (EVSIDS), which is a variation that yields identical decisions but avoids a costly loop over all variables to compute Equation (18). Rather than decaying the activity, the increment $\Delta$ is instead scaled up:

$$\Delta \leftarrow \frac{1}{\beta}\Delta. \tag{19}$$

The cumulative values of the activity scores then yield the same decisions. To incorporate our variable weights $w$ into this process, we simply modify Equation (17) by scaling the increment with the variable weight:

$$\text{activity}(x) \leftarrow \text{activity}(x) + w(x) \cdot \Delta. \tag{20}$$

This ensures that the total activity score of each variable is scaled by a factor of $w(x)$ at each step of the search while still preventing loops over all variables. We found that the runtime overhead of the additional multiplication in Equation (20) is negligible. We use the provided polarities $p(x)$ to initialize the polarity (or phase) of each variable. Note that we leave phase saving on, so this initial polarity may be overwritten by the solver in later search steps. We run all experiments without randomized decisions (rnd-freq = 0). We further set the parameter $K = 0.1$ to minimize solver restarts, which we found to improve performance on the three instance distributions considered in our experiments. Apart from this, we use the default parameters of Glucose.

MARCH

March (Heule et al., 2005; Heule & Van Maaren, 2006) is a DPLL-based solver that uses a branching heuristic based on look-ahead (Biere et al., 2021).[2] It is among the best-known solvers for purely random SAT instances. Look-ahead branching heuristics estimate how each variable's selection as a branching variable would affect the instance. In March, the scoring function SCORE(X) essentially quantifies how many new binary clauses would occur if $x$ is picked for branching in the current search step. Computing this score is relatively expensive when compared to activity-based approaches, and look-ahead solvers usually make fewer decisions per time. To decrease the cost of each branching step, March first applies a pre-selection step before each branching decision, where a reduced set of candidate variables is selected according to a second scoring function SCORE-PRESELECT($x$). This score aims to approximate the expected look-ahead score but is cheaper to compute. In the modified solver, we also apply the variable weight $w$ in pre-selection, i.e. the weighted scores $w(x) \cdot$ SCORE-PRESELECT($x$) are used to select the candidate variables. The ratio of pre-selected candidates is fixed at 10%. The same weights $w$ are then applied again to the actual look-ahead scores to obtain the branching variable. Afterwards, we use the given polarities $p$ in each branching to determine the sign of the branching literal. Aside from these changes, we run March in its default configuration.

---

[2]`https://github.com/marijnheule/march-SAT-solver`

---

**Algorithm 4** GRPO Training for SAT Solver Guidance

---

1: **Input:**
2:     Training formulas $\mathcal{F} = \{\phi_1, \ldots, \phi_N\}$
3:     Number of GRPO iterations $K \in \mathbb{N}$
4:     Number of samples per instance $M \in \mathbb{N}$
5:     Number of optimizer steps per GRPO iteration $S \in \mathbb{N}$
6:     Clip ratio $\epsilon \in (0, 1)$, KL penalty weight $\beta \geq 0$, learning rate $\eta > 0$
7: **Initialize:** Random weights $\theta_0$
8: **for** $k = 1, 2, \ldots, K$ **do**
9:     **for** $i = 1, 2, \ldots, N$ **do**
10:         **for** $j = 1, 2, \ldots, M$ **do**
11:             $\mathcal{W}_{i,j} \sim \pi_{\theta_{k-1}}(\phi_i)$
12:             $C_{i,j} \leftarrow \texttt{Cost}(\phi_i, \mathcal{W}_{i,j})$
13:             $R(\phi_i, \mathcal{W}_{i,j}) \leftarrow -C_{i,j}$
14:         **end for**
15:         $\mathbf{R}_i \leftarrow \{R(\phi_i, \mathcal{W}_{i,j}) \mid j \in \{1, \ldots, M\}\}$
16:         **for** $j = 1, 2, \ldots, M$ **do**
17:             $\hat{A}_{i,j} \leftarrow \frac{R(\phi_i, \mathcal{W}_{i,j}) - \text{mean}(\mathbf{R}_i)}{\text{std}(\mathbf{R}_i)}$
18:         **end for**
19:     **end for**
20:     $\theta \leftarrow \theta_{k-1}$
21:     **for** $s = 1, 2, \ldots, S$ **do**
22:         **for** $i = 1, 2, \ldots, N$ **do**
23:             **for** $j = 1, 2, \ldots, M$ **do**
24:                 $r_{i,j}(\theta) \leftarrow \frac{\pi_\theta(\mathcal{W}_{i,j} \mid \phi_i)}{\pi_{\theta_{k-1}}(\mathcal{W}_{i,j} \mid \phi_i)}$
25:             **end for**
26:             $\mathcal{L}_{\text{PPO}}(\theta \mid \phi_i) \leftarrow \frac{1}{M} \sum_j \left[ \min\left( r_{i,j}(\theta) \hat{A}_{i,j}, \text{clip}(r_{i,j}(\theta), 1-\epsilon, 1+\epsilon) \hat{A}_{i,j} \right) \right]$
27:             $\mathcal{L}(\theta \mid \phi_i) \leftarrow \mathcal{L}_{\text{PPO}}(\theta \mid \phi_i) - \beta \cdot \text{KL}\left( \pi_\theta(\phi_i), \pi_{\theta_{k-1}}(\phi_i) \right)$
28:         **end for**
29:         $\theta \leftarrow \theta + \eta \nabla_\theta \sum_i \mathcal{L}(\theta \mid \phi_i)$
30:     **end for**
31:     $\theta_k \leftarrow \theta$.
32: **end for**
33: **Output:** Final model weights $\theta_K$.

---

# B    EXPERIMENT DETAILS

## B.1    DATA

Table 2 provides full dataset statistics for all data distributions and splits. In the following, we provide further details on how each instance distribution is generated.

**Random 3SAT**    Uniformly random 3SAT instances are commonly used to benchmark SAT solvers. Here, each clause is sampled by choosing three distinct variables uniformly at random and negating each with a probability of 50%. Hard instances are known to occur when the number of clauses is around $m = 4.258n + 58.26n^{-\frac{2}{3}}$ where $n$ is the number of variables (Crawford & Auton, 1996). This is approximately the critical density where the instances transition from SAT to UNSAT. We define $3SAT(n)$ as the distribution of uniformly random 3SAT instances with $n$ variables and $\lceil 4.258n + 58.26n^{-\frac{2}{3}} \rceil$ clauses. For training, we use 20K instances sampled from $3SAT(200)$, which are filtered such that exactly 10K instances are SAT and UNSAT, respectively. Our test sets contain larger instances with $n \in \{300, 350, 400\}$, where we sample 200 instances for each size $n$.

**Graph Coloring**    Combinatorial problems on graphs are commonly solved by reducing them to Boolean SAT instances. Here, we consider the problem of finding a 3-coloring for Erdős-Rényi graphs. We define $3COL(n)$ as the distribution of SAT problems that are obtained by sampling an Erdős-Rényi graph with $n$ vertices and then encoding the problem of deciding 3-colorability as a SAT instance. We set the edge probability such that the expected vertex degree is $4.67$, which is approximately the critical density for 3-colorability where hard instances commonly occur (Zdeborová & Krząkała, 2007). We train on 20K instances sampled from $3COL(300)$. Again, these are filtered such that exactly 10K instances are SAT and UNSAT, respectively. Our test sets consist of larger problems with $n \in \{400, 500, 600\}$.

**Cryptographic**    Hard, structured SAT problems commonly arise in the context of cryptoanalysis, for example, for SAT-based decryption attacks (Soos et al., 2009). To generate data in this domain, we use Grain-of-Salt (Soos, 2010) to generate SAT instances for decrypting stream ciphers. We define $CRYPTO(n)$ as the distribution of SAT instances generated for decrypting the HiTag2 cipher (Courtois et al., 2009) with $n$ given help bits. We use the recommended generation parameters (`-outputs 56 -base-shift 8 -karnaugh 8`). Note that these instances are harder for smaller values of $n$ and are mostly UNSAT. We train on 20K instances from $CRYPTO(22)$ and test on harder problems with $n \in \{20, 15, 10\}$.

For each of these three instance classes we formally define the corresponding training distribution $\Omega$ from Equation (6) as the uniform distribution over the set of training instances.

Table 2: Dataset Statistics

| | Distribution | Split | Number | #SAT | #UNSAT | $|\mathrm{Var}(\phi)|$ | | | $|\mathrm{Cls}(\phi)|$ | | |
| | | | | | | mean | min | max | mean | min | max |
|---|---|---|---|---|---|---|---|---|---|---|---|
| 0 | 3SAT(200) | Train | 20,000 | 10,000 | 10,000 | 200.00 | 200 | 200 | 853.00 | 853 | 853 |
| 1 | 3SAT(200) | Val | 200 | 100 | 100 | 200.00 | 200 | 200 | 853.00 | 853 | 853 |
| 2 | 3SAT(300) | Test | 200 | 103 | 97 | 300.00 | 300 | 300 | 1,278.00 | 1,278 | 1,278 |
| 3 | 3SAT(350) | Test | 200 | 108 | 92 | 350.00 | 350 | 350 | 1,491.00 | 1,491 | 1,491 |
| 4 | 3SAT(400) | Test | 200 | 89 | 111 | 400.00 | 400 | 400 | 1,704.00 | 1,704 | 1,704 |
| 5 | 3Col(300) | Train | 20,000 | 10,000 | 10,000 | 900.00 | 900 | 900 | 3,284.75 | 3,009 | 3,618 |
| 6 | 3Col(300) | Val | 200 | 100 | 100 | 900.00 | 900 | 900 | 3,288.63 | 3,042 | 3,489 |
| 7 | 3Col(400) | Test | 200 | 77 | 123 | 1,200.00 | 1,200 | 1,200 | 4,392.31 | 4,144 | 4,603 |
| 8 | 3Col(500) | Test | 200 | 91 | 108 | 1,500.00 | 1,500 | 1,500 | 5,488.56 | 5,216 | 5,750 |
| 9 | 3Col(600) | Test | 200 | 87 | 113 | 1,800.00 | 1,800 | 1,800 | 6,597.24 | 6,306 | 6,861 |
| 10 | Crypto(22) | Train | 20,000 | 0 | 20,000 | 529.41 | 518 | 544 | 8,420.71 | 7,669 | 9,453 |
| 11 | Crypto(22) | Val | 200 | 0 | 200 | 529.24 | 523 | 537 | 8,413.41 | 7,937 | 9,075 |
| 12 | Crypto(20) | Test | 100 | 0 | 100 | 533.43 | 526 | 544 | 8,767.57 | 8,182 | 9,309 |
| 13 | Crypto(15) | Test | 100 | 0 | 100 | 542.89 | 537 | 552 | 9,622.04 | 9,129 | 10,321 |
| 14 | Crypto(10) | Test | 100 | 0 | 100 | 550.99 | 544 | 568 | 10,497.63 | 9,947 | 11,528 |

## B.2 Hyperparameters

Table 3 provides an overview of all RLAF training runs from our main experiments. We tuned the learning rate in $\eta \in \{0.0001, 0.00005, 0.00001\}$ and schedule it to warm up over the first 5 GRPO iterations. After warm up the the learning rate stays constant throughout training. The clip ratio was tuned in $\epsilon \in \{0.1, 0.2\}$ and the KL-penalty $\beta \in \{0.1, 1.0\}$. All other hyperparameters were given constant default values, which we found to be stable based on preliminary experiments. Each training run uses a machine equipped with a single H100 GPU, an Intel Xeon 8468 CPU with 48 cores, and 128GB of RAM. The total runtime of all training runs is between 24h and 48h.

Table 3: Hyperparameters

|  | Glucose | | | March | | |
|---|---|---|---|---|---|---|
|  | 3Sat | 3Col | Crypto | 3Sat | 3Col | Crypto |
| $K$ | 2000 | 2000 | 2000 | 2000 | 2000 | 2000 |
| $M$ | 40 | 40 | 40 | 40 | 40 | 40 |
| $N$ | 100 | 100 | 100 | 100 | 100 | 100 |
| $S$ | 50 | 50 | 50 | 50 | 50 | 50 |
| $\sigma^w$ | 0.1 | 0.1 | 0.1 | 0.1 | 0.1 | 0.1 |
| clip ratio $\epsilon$ | 0.2 | 0.2 | 0.2 | 0.1 | 0.2 | 0.2 |
| KL-penalty $\beta$ | 0.1 | 1.0 | 0.1 | 1.0 | 0.1 | 0.1 |
| batch size | 20 | 20 | 20 | 20 | 20 | 20 |
| learning rate $\eta$ | 0.0001 | 0.00005 | 0.00005 | 0.00005 | 0.00001 | 0.0001 |
| weight decay | 0.0 | 0.0 | 0.0 | 0.0 | 0.0 | 0.0 |
| hidden dim $d$ | 256 | 256 | 256 | 256 | 256 | 256 |
| model depth $L$ | 10 | 10 | 10 | 10 | 10 | 10 |

## B.3 Training

Figure 6 provides the learning curves for the 6 RLAF-trained models in our main experiments. For all models, the cost decreases throughout training. We found that training with the March base solver tends to yield noisier training, particularly on 3Sat instances, where the policy does not improve further after 700 GRPO iterations. Exploring effective strategies for reducing this noise remains future work. Nonetheless, we can learn guidance policies that reduce the solver cost of both base solvers across all three problem instances.

We further note that, in principle, any RL method based on policy gradients can be applied to learn a GNN policy for our MDP formulation. During development we briefly experimented with using DPO Rafailov et al. (2023) instead of GRPO and observed that training with DPO also converges effectively. However, the learned policies where less effective on test problems compared to GRPO-learned policies, which is we we opted for GRPO as our default training methodology.

## B.4 Extended Results

In Table 4 we provide the results from our main experiments and additionally report the mean wall-clock runtime of the GNN forward pass. For all instance distributions, this GNN overhead is between 0.02 and 0.1 seconds, which is negligible when compared to Sat solver runtimes on non-trivial instances. However, we note that classical Sat solvers commonly perform over $10^4$ branching decisions per second. In a setting where every guided branching decision requires a separate forward pass, as in prior RL-based work (Kurin et al., 2020; Cameron et al., 2024), it is therefore not possible to guide every branching decision without incurring a massive runtime overhead. Our one-shot setup avoids this problem as it incorporates multiplicative weights obtained in a single GNN pass in every branching decision with minimal runtime overhead.

Figure 7 further provides survival plots that plot the cumulative number of solved problems against the wall-clock runtime on each instance distribution. With the exception of March on Unsat 3Sat instances, RLAF constantly improves the speed of the guided base solver.

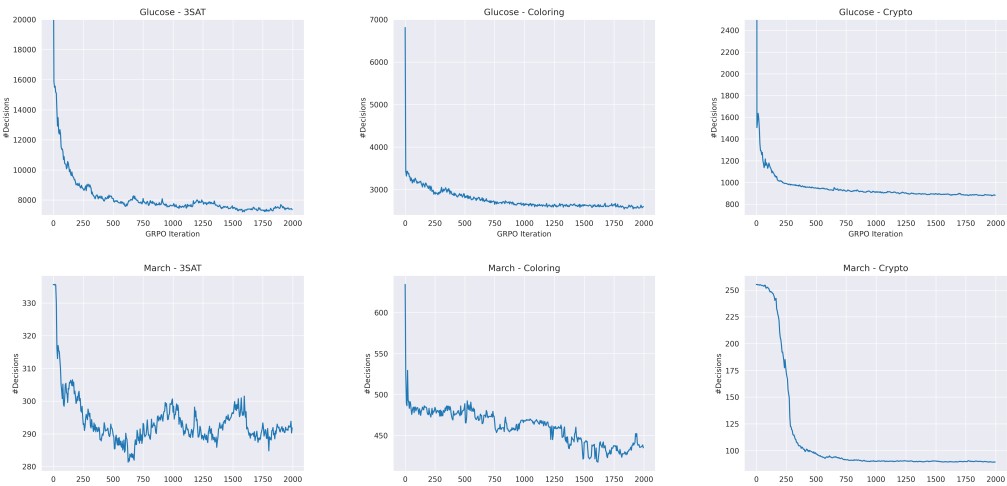

Figure 6: GRPO training curves of the RLAF models from our main experiment. We plot the mean number of decisions on the validation set against the GRPO iteration.

Table 4: Full results on test instances, including the main time spent for the GNN forward pass. All metrics are averaged across the respective test sets. The mean number of decisions is rounded to the nearest whole number. For results with RLAF, we include the time required for the GNN forward pass in the total runtime.

| Data | | Glucose | | Glucose + RLAF | | | March | | March + RLAF | | |
|---|---|---|---|---|---|---|---|---|---|---|---|
| Distribution | Result | Decisions | Time (s) | Decisions | Time (s) | GPU time (s) | Decisions | Time (s) | Decisions | Time (s) | GPU time (s) |
| 3SAT(300) | SAT | 341,418 | 6.67 | 121,184 | 1.85 | 0.0210 | 2,893 | 0.25 | 2,389 | 0.23 | 0.0205 |
| 3SAT(300) | UNSAT | 725,812 | 15.49 | 508,676 | 8.21 | 0.0209 | 11,783 | 1.01 | 11,757 | 1.04 | 0.0205 |
| 3SAT(350) | SAT | 1,568,289 | 48.76 | 805,035 | 18.88 | 0.0238 | 16,546 | 1.64 | 11,702 | 1.19 | 0.0233 |
| 3SAT(350) | UNSAT | 3,628,268 | 132.28 | 3,136,552 | 82.84 | 0.0237 | 52,287 | 5.14 | 51,846 | 5.16 | 0.0233 |
| 3SAT(400) | SAT | 9,638,668 | 598.35 | 4,447,304 | 186.70 | 0.0265 | 64,296 | 7.27 | 47,992 | 5.51 | 0.0272 |
| 3SAT(400) | UNSAT | 22,130,692 | 1,895.62 | 20,808,043 | 1,112.71 | 0.0265 | 245,064 | 27.49 | 242,499 | 27.51 | 0.0270 |
| 3Col(400) | SAT | 15,519 | 0.36 | 6,988 | 0.22 | 0.0662 | 926 | 0.22 | 598 | 0.21 | 0.0661 |
| 3Col(400) | UNSAT | 70,692 | 1.99 | 34,920 | 0.81 | 0.0662 | 10,563 | 2.61 | 5,954 | 1.57 | 0.0661 |
| 3Col(500) | SAT | 82,758 | 2.61 | 35,901 | 1.05 | 0.0853 | 7,689 | 2.55 | 4,754 | 1.68 | 0.0848 |
| 3Col(500) | UNSAT | 460,881 | 17.47 | 363,278 | 12.24 | 0.0855 | 100,811 | 33.34 | 60,321 | 20.52 | 0.0849 |
| 3Col(600) | SAT | 606,598 | 25.03 | 339,378 | 11.59 | 0.0984 | 63,512 | 27.16 | 42,862 | 18.71 | 0.0988 |
| 3Col(600) | UNSAT | 3,092,344 | 193.96 | 2,811,133 | 155.23 | 0.0984 | 754,720 | 313.57 | 461,639 | 197.12 | 0.0990 |
| Crypto(20) | UNSAT | 51,294 | 1.16 | 3,541 | 0.15 | 0.0974 | 1,203 | 0.82 | 390 | 0.41 | 0.0973 |
| Crypto(15) | UNSAT | 225,447 | 5.74 | 64,150 | 1.40 | 0.1075 | 52,282 | 34.56 | 8,257 | 6.40 | 0.1073 |
| Crypto(10) | UNSAT | 3,753,850 | 162.45 | 1,520,075 | 64.95 | 0.1148 | 679,864 | 467.38 | 230,905 | 169.22 | 0.1174 |

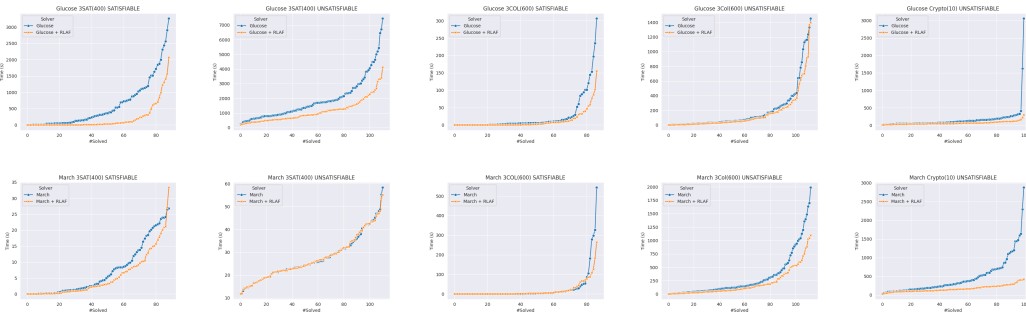

Figure 7: Survival plots of our main results for different instance distributions.

### B.5 Supervised Unsat-Core and Backbone Prediction

For UNSAT-core and backbone prediction in Section 3.2, we train supervised GNN models that are identical in architecture and size to the models used for our RLAF-based policies. The used hyperparameters are specified in Table 5. In the following, we provide a detailed description of how these models are trained and evaluated.

#### Unsat-Core

Selsam & Bjørner (2019) propose to train supervised models that predict the UNSAT-core membership of variables and then use the model prediction to guide branching heuristics. Following their methodology, we phrase the task of predicting whether or not a variable occurs in an UNSAT-core as a variable-level binary classification task and train a GNN for this problem in a supervised manner using a standard cross-entropy loss. The ground-truth on training and validation instances is computed by extracting the cores from DRAT UNSAT proofs generated by the CaDiCaL Biere et al. (2024) solver. Note that these cores are not minimal, as computing such would not be feasible. As discussed in Section 3.2, the extracted cores on our unsatisfiable 3SAT training instances contain all variables for almost all instances and are therefore not a meaningful training target. We therefore only train UNSAT-core prediction models for 3COL and CRYPTO. We train a separate model for each distribution and restrict training to the unsatisfiable instances.

Note that Selsam & Bjørner (2019) integrate their prediction by periodically resetting the VSIDS scores of the guided CDCL-solver to prediction logits of the GNN. This requires careful tuning of the reset frequency. It is also specific to solvers based on the VSIDS heuristic and would, for example, not be applicable to the March solver. Furthermore, in later ablation experiments, Selsam & Bjørner (2019) report that the performance improvement obtained with a trained GNN is barely distinguishable from when an untrained, randomly initialized model is used, further questioning the effectiveness of guiding solvers with this strategy. To facilitate a direct and fair comparison with RLAF-trained policies, we instead combine the UNSAT-core predictions with our own solver guidance based on multiplicative weights. For a variable $x$, let $p_{\text{core}}(x)$ be the predicted probability of $x$ being in an UNSAT-core according to the trained GNN model. Then we transform these probabilities to variable weights through the following transformation:

$$w(x) = 1 + \alpha \cdot p_{\text{core}}(x). \tag{21}$$

Here, $\alpha \geq 0$ is a parameter that determines how the variable weight scales with the raw model predictions. For this experiment, we found weights of $w(x) \geq 1$ to perform better, hence the offset of 1 in Equation (21). The value of $\alpha$ is tuneed on the corresponding validation dataset in the range $\{10^{-4}, 10^{-3}, 10^{-2}, 10^{-1}, 10^0, 10^1, 10^2, 10^3, 10^4\}$. We tune $\alpha$ separately for both Glucose and March. The polarities are simply set to $p(x) = 1$ as the prediction of UNSAT-core membership has no clear implication for the sign of the branching literal. Using this methodology, we found that the UNSAT-core predictions can significantly accelerate both base solvers, although by a smaller margin than RLAF-trained policies.

#### Backbone

Wang et al. (2024) suggests using the backbone membership of literals as a supervised training target and then setting variable polarities using the model predictions. We follow their methodology and train a GNN on the literal-level binary classification task using cross-entropy loss. As discussed in Section 3.2, we only train a model for the 3SAT instances and only use the satisfiable problems for training. The backbone of coloring problems is always empty due to the permutation symmetry of the colors, and some distributions, such as CRYPTO, predominantly consist of UNSAT instances.

When evaluating, we set the polarity of a variable $x$ as $p(x) = 0$ if $p_{\text{backbone}}(\neg x) > p_{\text{backbone}}(x)$ and $p(x) = 1$ otherwise. Here, $p_{\text{backbone}}(\ell)$ is the predicted probability of literal $\ell$ belonging to the backbone. We further assign variable weights $w(x)$ under the assumption that correctly assigning backbone literals in early search steps positively affects the runtime. To this end, we apply the transformation from Equation (21) to the mean backbone probability $\overline{p}_{\text{backbone}}(x) = 0.5(p_{\text{backbone}}(\neg x) + p_{\text{backbone}}(x))$ to obtain a weight for each variable. Again, we tune the transformation parameter $\alpha$ for both base solvers on the validation set.

Table 5: Hyperparameters of the supervised models.

|  | 3SAT | 3COL | CRYPTO |
|---|---|---|---|
| batch size | 50 | 50 | 50 |
| learning rate $\eta$ | 0.0001 | 0.0001 | 0.0001 |
| weight decay | 0.1 | 0.1 | 0.1 |
| epochs | 200 | 200 | 200 |
| hidden dim $d$ | 256 | 256 | 256 |
| model depth $L$ | 10 | 10 | 10 |
| $\alpha$ Glucose | $10^1$ | $10^3$ | $10^{-2}$ |
| $\alpha$ March | $10^{-2}$ | $10^{-2}$ | $10^1$ |

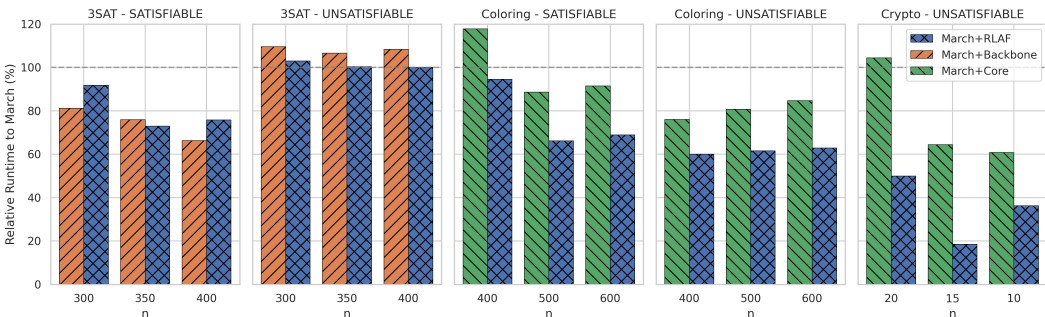

Figure 8: Runtimes relative to the base solver March for RLAF and supervised approaches based on Backbones and UNSAT cores. **Less is better**. We include the time required for the GNN forward pass in the runtime.

### SUPERVISED COMPARISON WITH MARCH

In Figure 8, we further provide the comparison with supervised baselines from Section 3.2 for the March base solver. On satisfiable 3SAT problems, our RLAF-trained policy and the guidance based on backbone prediction are roughly on par. However, on unsatisfiable 3SAT problems we found that backbone-based guidance *increases* the solver's runtime be approximately 10%. Backbone predictions are therefore not a useful guidance signal on this instance type when working with a strong base solver, such as March. Our RLAF-based policy does not share this problem. On the 3COL and CRYPTO distributions, the RLAF-trained policy consistently outperforms the guidance based on UNSAT core prediction, as for the Glucose base solver.

### B.6 COMPARISON TO GRAPH-Q-SAT

A central design choice in RLAF is *one-shot* solver guidance: a single GNN forward pass assigns multiplicative variable weights and polarities that are reused at every branching decision with negligible overhead (Section B.4). Prior RL/GNN methods, such as Graph-Q-SAT (Kurin et al., 2020), couple the GNN tightly to the solver by invoking it at each guided decision, which can introduce substantial runtime costs. Graph-Q-SAT proposes an RL formulation where every solver decision corresponds to one action of an MDP where a GNN is tasked with selecting the next branching variable. The model is then trained with standard Q-learning techniques. Note that Q-learning is not applicable for our RL formulation, which is based on a large, combinatorial, and partially continuous action space that is more suitable for policy-gradient methods such as GRPO.

To facilitate a comparison with RLAF within this landscape, we include an evaluation on 3SAT instances from SATLIB. As the public Graph-Q-SAT codebase did not run in our environment, we compare against the *reported* Graph-Q-SAT MRIR from Kurin et al. (2020) on the same SAT instances. The base solver is MiniSAT (Eén & Sörensson, 2003), which we extended with RLAF guidance in a manner identical to the guided Glucose solver used in our main experiments. Training uses 800 satisfiable 50-variable 3SAT instances from SATLIB. Evaluation covers 3SAT problems

Table 6: SATLIB comparison with the MiniSAT base solver. We report MRIR metric for the compared models (higher is better). We also provide the mean decision counts of the unguided MiniSAT solver as a reference.

| Dataset | MiniSAT #Decisions | Graph-Q-SAT | RLAF-SATLIB | RLAF-Main |
|---|---|---|---|---|
| SAT 100–430 | 286 | 3.94 | 3.14 | 3.58 |
| SAT 250–1065 | 76,120 | 3.91 | 11.03 | 7.33 |
| UNSAT 100–430 | 596 | 2.24 | 1.56 | 2.03 |
| UNSAT 250–1065 | 182,799 | 1.54 | 1.30 | 1.94 |

with $n \in \{100, 250\}$ variables, including both SAT and UNSAT splits. We report *MRIR* (Mean Relative Improvement in Runtime) as defined by Kurin et al. (2020), averaging over five independent RLAF models (denoted as *RLAF-SATLIB*). As a reference, we also evaluate the policy trained for our main experiments on mixed SAT/UNSAT data from 3SAT(250), which we denote as *RLAF-Main*. Note that this policy was learned with the Glucose solver but is also capable of guiding the closely related MiniSAT solver effectively.

All learned policies accelerate MiniSAT. On small 100-variable SAT instances, the reported Graph-Q-SAT MRIR exceeds RLAF-SATLIB. On larger 250-variable SAT instances, RLAF-SATLIB attains a $\sim 3\times$ higher MRIR than the reported Graph-Q-SAT, indicating strong size generalization for one-shot guidance. For UNSAT, RLAF-SATLIB underperforms the reported Graph-Q-SAT MRIR, consistent with training solely on SAT instances. The solver guidance from our main experiments (RLAF-Main) achieves the highest MRIR on the 250-variable UNSAT set, which aligns with its training on mixed SAT/UNSAT data and the consistent solver guidance on long search trajectories with >100K branching decisions.

These results support two observations. First, a one-shot multiplicative-weights interface can provide persistent guidance throughout the search while avoiding per-decision GNN calls, achieving clear gains on larger SAT instances. Second, policy-gradient training over a joint, partially continuous action space is effective for learning guidance that scales in problem size and can transfer across EVSIDS-based solvers. The incurred wall-clock overhead is only a single forward pass (Section B.4), enabling guidance of *all* decisions without prohibitive inference costs.

## B.7 Impact of Weights and Polarities

Our learned policies provide both variable weights and polarities to guide the solver. Here, we aim to measure the individual impact of these model outputs. To this end, we evaluate the models from our main experiment in a modified setting where only the weights or the polarities are provided to the guided solver. We create a setting where only the variable weights are used by setting all polarities to 1. In the inverse setting, we set all variable weights to 1 and only use the predicted polarities.

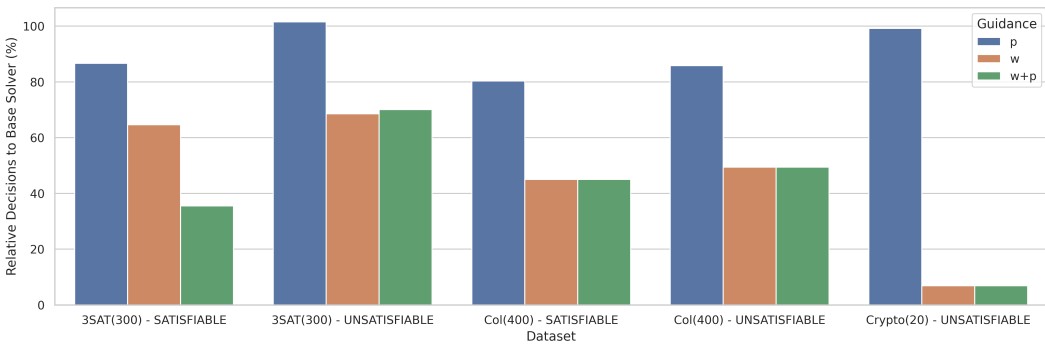

Figure 9: Relative mean number of decisions compared to the Glucose base solver. **Less is better**. We compare 3 guidance settings: Using polarities only (p), using weights only (w), and using both weights and polarities (p+w).

Figure 9 provides the test results in both settings as well as the default setting where both weights and polarities are used. Overall, we observe that variable weights account for most of the performance gains. On satisfiable 3SAT problems, the polarities do play a significant role in accelerating the solver, as using both weights and polarities performs significantly better than using either separately. On the remaining instance distributions, guidance with weights only already achieves the same speed-up as using both weights and polarities. Conceptually, well-chosen variable polarities are most impactful on satisfiable problems with non-empty backbones, and our empirical results reflect this. Variable weights, on the other hand, are ubiquitously useful across instance distributions.

## B.8 IMPACT OF THE GNN DEPTH

We further study how the depth of the underlying GNN affects the quality of the learned guidance. Recall that this depth was set to 10 in our main experiments. We train two additional policies to guide Glucose on 3SAT with GNN depths of 3 and 5, respectively. Figure 10a provides the mean number of decisions achieved by each policy on the 3SAT(300) test set. We observe that deeper GNNs yield more effective guidance. These results demonstrate that the learned policies effectively exploit the model's depth and incorporate non-trivial structural graph features that are not captured by shallow GNNs. These results also suggest that scaling up the GNN model in the future may further improve performance.

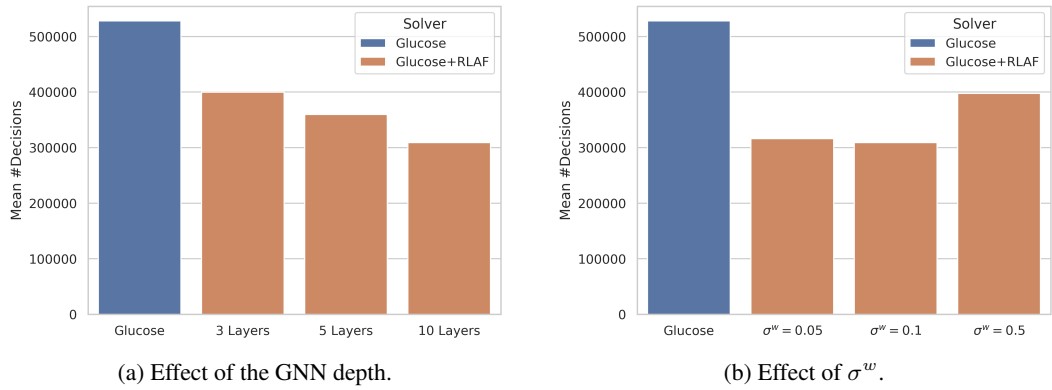

(a) Effect of the GNN depth.         (b) Effect of $\sigma^w$.

Figure 10: Ablation experiments on the effects of GNN depth and $\sigma^\omega$ parameter. We report the mean number of decisions achieved on the 3SAT(300) test set (SAT and UNSAT).

## B.9 IMPACT OF $\sigma^w$

We further provide data on the impact of the $\sigma^w$ hyperparameter, which sets the variance of the LogNormal variable weight distributions. By default, we use a fixed value of $\sigma^w = 0.1$ in our main experiments, which was determined in preliminary tuning runs. To probe the effect of using a wider range of values for $\sigma^w$, we train two additional policies to guide Glucose on 3SAT with $\sigma^w = 0.05$ and $\sigma^w = 0.5$. Both of these additional policies converged stably during training. Figure 10b compares the test results of these policies. We observe that the three policies all successfully accelerate the base solver. The policy trained with $\sigma^w = 0.05$ performs nearly identically to our default of $\sigma^w = 0.1$, while training with a higher value of $\sigma^w = 0.5$ does cause some performance degradation. This suggests that successfully training a policy with RLAF does not require the specific choice of $\sigma^w = 0.1$. However, as the test-time performance is somewhat sensitive to the choice of this hyperparameter, it should be appropriately tuned for a given application of RLAF.

