# OpenReview forum: "Learning from Algorithm Feedback: One-Shot SAT Solver Guidance with GNNs"
_ICLR.cc/2026/Conference — ICLR 2026 Poster_

### Official Review · Reviewer_j7Rf · 2025-10-30

**Soundness:** 3
**Presentation:** 3
**Contribution:** 3
**Rating:** 6
**Confidence:** 2

**Summary:**

This paper introduces Reinforcement Learning from Algorithm Feedback (RLAF), a paradigm for training Graph Neural Networks to guide SAT solver branching heuristics through a one-shot mechanism where a single GNN forward pass assigns multiplicative variable weights w(x) and polarities p(x) that persistently influence all branching decisions. The approach modifies existing solvers by scaling their native scoring functions with learned weights, formulating weight selection as a single-step MDP trained with GRPO policy-gradient methods using only solver cost as reward, eliminating expert supervision or predefined properties like UNSAT cores. Training on easy instances with 2000 GRPO iterations sampling M=40 parameterizations across N=100 instances, the policies generalize to larger problems, achieving 69% runtime reduction on 400-variable 3SAT, 2× speedup on 600-vertex graph coloring, and 3× improvement on cryptographic instances. The approach consistently outperforms supervised baselines predicting handcrafted variable properties, with learned weights from different solvers showing high correlation (r=0.73-0.85), suggesting solver-agnostic structural understanding. The one-shot design incurs minimal overhead (0.02-0.1s) versus prior RL methods requiring one GNN pass per decision, enabling practical guidance throughout trajectories exceeding 100K decisions.

**Strengths:**

1. The paper is well-written with good illustration and easy to follow. The paper provides clear algorithmic descriptions (Algorithms 1-3) showing precisely how variable weights are injected into branching heuristics, with Figure 2 effectively illustrating the complete pipeline from graph representation through GNN processing to solver guidance. The background section carefully explains both SAT solving fundamentals and RL formulations, making the paper accessible to readers from either community. Experimental setup is thoroughly documented with comprehensive hyperparameter tables and detailed data generation procedures in the appendix.
2. The learned policies achieve substantial speedups across diverse problem distributions, including 69% runtime reduction on satisfiable 3SAT(400) instances and more than 2× acceleration on graph coloring and cryptographic problems, while generalizing to instances significantly larger than those seen during training. RLAF consistently outperforms supervised baselines that predict handcrafted heuristics like UNSAT cores and backbones, demonstrating that direct RL optimization yields more effective guidance. The one-shot design incurs negligible overhead (0.02-0.1 seconds) compared to solver runtime, making the approach practically viable.

**Weaknesses:**

This paper is in general a good paper to me, but I am not an expert in this field. Here I point out some suggestions which I hope could strengthen the paper further.

While the paper demonstrates strong results on three well-studied problem classes, the evaluation could be strengthened by including more diverse and realistic industrial SAT benchmarks from domains like hardware verification, software testing, or planning problems from SAT competitions.

For example, the choice of LogNormal distribution for variable weights (Equation 2) with fixed standard deviation σ_w=0.1 is mentioned to perform best in "preliminary experiments," but no systematic comparison with alternative distributions (e.g., Gamma, truncated Normal) or analysis of sensitivity to σ_w is provided.

**Questions:**

See the weakness part.

---

> ### Author Response · Authors · 2025-11-21
> **Response to Reviewer j7Rf**
>
> We thank the reviewer for their positive assessment, specifically for highlighting the the effectiveness of the one-shot design and the strong empirical results compared to supervised baselines. We address your suggestions below:
>
> **Industrial Benchmarks**
>
> We agree that evaluating on industrial benchmarks is a crucial direction for GNN-for-SAT research. However, as noted in Section 4, this work serves as a methodological demonstration of the RLAF paradigm using a prototype-scale implementation. Large-scale industrial instances (e.g., from SAT competitions) impose distinct scalability and memory challenges for GNNs that differ from the learning challenges addressed here. We view the engineering required to scale RLAF to large industrial instances as the immediate next step for this line of research.
>
> **LogNormal Distribution and Sensitivity Analysis**
> Thank you for raising this point. We have updated the paper to address your concerns:
>
> * **Sensitivity Analysis:** We have added a sensitivity analysis of the standard deviation parameter $\\sigma^w$​ to the revised paper (Appendix B.9). Our results indicate that RLAF training converges effectively across a range of values for $\\sigma^w$, though tuning the parameter is beneficial for maximizing the final solver speedup.
> * **Choice of Distribution:** We selected the LogNormal distribution because it is a standard, numerically stable choice for modeling unimodal distributions over positive real numbers, which is required for multiplicative weights. While we experimented with alternatives (such as Truncated Normal and Poisson distributions) during early development, we found LogNormal to be simpler and more robust for this application. We have clarified the rationale for this design choice in the text (Section 2.3) and believe a comprehensive ablation of alternative distribution types falls outside the scope of this initial study.

---

### Official Review · Reviewer_gUdh · 2025-10-31

**Soundness:** 3
**Presentation:** 3
**Contribution:** 3
**Rating:** 6
**Confidence:** 2

**Summary:**

This work introduces Reinforcement Learning from Algorithm Feedback (RLAF), a paradigm for learning to guide SAT solver branching heuristics using Graph Neural Networks (GNNs).

The authors demonstrate that RLAF-trained policies can reduce the mean solve time of various base solvers across a diverse set of SAT problems.

Many branching heuristics-based methods are implemented by starting with selecting a variable
\hat{x} = argmax_x Score(x)
that maximizes some function Score.

The authors modify this to incorporate additional variable weights w for the given input formula:
\hat{x} = argmax_x w(x) · Score(x)

This way, we can inject prior knowledge of variable importance into the solver’s branching decisions.
In addition to the weights w(x), polarity p(x) can also be assigned to each variable x. When x is chosen as a decision variable, the polarity determines which value is assigned to x first.

The authors map input formula Phi to a graph representation (“Literal-Clause Graph”) that captures the instance’s structure. This graph is processed by a GNN that extracts structural information.
The output of the GNN [mu(x), rho(x)] is used to parameterize variable-wise weight and polarity distributions.
The models are trained by RL: the input formula Phi is the state, and a variable parameterization is the action.
Once the action is taken, the environment transitions immediately to a terminal state, providing a reward R(Phi, W) = −Cost(Phi, W).

**Strengths:**

•	The paper is well written.
•	It investigates how RL-trained Graph Neural Networks (GNNs) can improve the branching heuristics of SAT solvers.
•	The authors modify existing DPLL-based backtracking SAT solvers to incorporate external variable weights into their branching heuristics.
•	The results demonstrate that, after training, the learned policies generalize well to significantly larger and more challenging problems.
•	I particularly appreciate the inclusion of the graph coloring and cryptographic experiments.

**Weaknesses:**

The work is demonstrated only on small-scale examples, with all training performed on machines equipped with a single multi-core CPU and one GPU. I wish the paper included more numerical experiments on larger-scale problems.

**Questions:**

•	How would the results change if other RL methods were used instead of GRPO?
•	How does the training time scale with problem size? How much faster is training on a GPU compared to a CPU?
•	What are the main challenges (if any) in developing a distributed version?

---

> ### Author Response · Authors · 2025-11-21
> **Response to Reviewer gUdh**
>
> We thank the reviewer for the insightful feedback and positive assessment.
>
> We agree that our current experiments are a "small-scale demonstration" constrained by our single-machine training setup. While we are encouraged that the learned policies consistently generalize to larger and harder test problems, we agree that demonstrating the approach on larger-scale training instances is a critical next step. As discussed in Section 4, we believe the path forward is to leverage distributed computing to scale up the collection of solver feedback.
>
> **How would the results change if other RL methods were used instead of GRPO?**
>
> In principle, other policy gradient methods can be applied to learn a policy for the underlying MDP. While developing the method, we explored the option of using DPO \[1\] instead of GRPO. We observed stable training convergence with DPO as well, but the learned policies achieved less solver acceleration than those trained with GRPO, which is why we opted for GRPO as our default. We have added a note on this observation to the paper (Appendix B.3).
>
> **How does the training time scale with problem size? How much faster is training on a GPU compared to a CPU?**
>
> The training time of our online RL setup scales proportionally to the solver's runtime on the training instances. In the worst case, this may of course be exponential in the instance size, which is why our training uses instance distributions with comparatively small problems. This training setup assumes that the GNN can learn a policy that effectively generalizes to larger instances after training, which is indeed the case in our experiments.
>
> Compared to large neural networks, such as LLMs, our GNN is relatively lightweight with just a few million parameters. A single GPU is therefore fully sufficient for running training on our current instance sizes. However, CPU-based training of the GNN would be prohibitively slow, as GNNs still require large matrix multiplications that CPUs are not optimized for. We have not explicitly benchmarked this setting, but it would likely be impractical.
>
> **What are the main challenges (if any) in developing a distributed version?**
>
> Setting up a distributed version would mainly be engineering work. It would require integrating the submission of distributed jobs to a suitable CPU-based compute cluster into the training loop. The details of this would depend on the details of the cluster (local slurm cluster, cloud resources, etc.). This would, of course, first require allocating such compute resources for this purpose. Scaling to larger problems may also require expanding the GNN training to a multi-GPU setup, as larger training graphs may exceed the available GPU memory during training.
>
>
> #### \[1\] Rafailov, Rafael, et al. "Direct preference optimization: Your language model is secretly a reward model." *Advances in neural information processing systems* 36 (2023): 53728-53741.

---

### Official Review · Reviewer_KMQK · 2025-11-01

**Soundness:** 4
**Presentation:** 4
**Contribution:** 4
**Rating:** 8
**Confidence:** 4

**Summary:**

The paper presents a novel method to combine (graph) neural networks and
a SAT solver. They use the network to predict an "importance" of each
variable, and then combine this value with SAT solver's inner variable
evaluation. The approach uses curriculum learning to train the NN with
RL (GRPO) on easy instance in one of three categories, and show time
improvement on harder problems, compared to the base solver as well as
other approaches.

**Strengths:**

* The use of a NN once before the solver doesn't slow down the solver,
   and directly relates to SOTA methods for SAT solving.
* The article is well written, including discussing the downsides

**Weaknesses:**

* Using RL forces the network to be trained only on small SAT instances,
   so it was not attempted on serious problems.

**Questions:**

-

---

> ### Author Response · Authors · 2025-11-21
> **Response to Reviewer KMQK**
>
> We thank the reviewer for the very positive assessment.
>
> We agree that the restriction to comparatively small training instances remains the main weakness of our online RL setup. However, we are encouraged that the learned policies consistently generalize well to significantly harder and larger test problems after training. We believe the clear path forward is to scale up solver feedback collection using distributed computing, enabling training on much larger and more complex problems.

---

### Official Review · Reviewer_CgoS · 2025-11-03

**Soundness:** 4
**Presentation:** 4
**Contribution:** 4
**Rating:** 6
**Confidence:** 3

**Summary:**

This paper introduces Reinforcement Learning from Algorithm Feedback (RLAF), a novel paradigm for accelerating SAT solvers by training a Graph Neural Network (GNN) to guide their branching heuristics. Departing from traditional approaches that rely on expert-crafted rules or supervised labels, RLAF formulates solver guidance as a one-shot reinforcement learning problem. Instead of intervening at each decision step, the GNN performs a single pass at the beginning of the search to assign an "importance score" (weight) and a "starting guess" (polarity) to each variable.
These predictions are integrated directly into the branching heuristics of existing solvers. The GNN policy is trained end-to-end using policy-gradient methods, such as GRPO, with the objective of minimizing the solver’s computational cost. The solver's own performance serves as the reward signal, creating a self-improving feedback loop that obviates the need for expert supervision. The experimental evaluation demonstrates consistent speedups across random 3-SAT, graph coloring, and cryptographic SAT problems. Notably, models trained on smaller instances generalize effectively to significantly larger and more difficult unseen problems.

**Strengths:**

-	The RLAF paradigm is a novel and effective concept. It learns by using the solver's own performance, i.e., its computational cost, as a reward signal. This help in eliminating the need for costly human labelling or hand-curated features (GNN can train itself).
-	The GNN runs just once per problem to generate a set of hints (weights and polarities) that guide the entire search. It resolves the computational bottleneck of prior RL/GNN methods that required per-decision GNN calls, which often made them slower than the original solver .
-	The proposed mechanism for feeding variable weights and polarities into existing branching heuristics is generic, which allows the learned policies to be applied to different base solvers.
-	Strong Empirical Results: The experimental evaluation is comprehensive and interesting. The authors demonstrate substantial speedups across 3 distinct SAT problem distributions and two different base solvers. It also generalizes well to larger and harder unseen instances. RLAF-trained policies also consistently outperforms over supervised approaches.
-	Clarity and Well-Structured Paper: The paper is clearly structured and well-written. Figures (especially Figs. 2–5) effectively convey key ideas. Mathematical formulations (e.g., the RL setup, Eqns. 6–10) are precise. Overall, its easy to follow.
-	Detailed appendices include architecture, solver modification details, and training algorithms. Reproducibility is well-handled.

**Weaknesses:**

The primary limitations of the work, which the authors are commendably upfront about, relate to the scalability of training and the depth of the methodological analysis. The most significant constraint is the computationally intensive nature of the RL training loop. This currently confines the approach to smaller problem instances, leaving its effectiveness on industrial-scale SAT problems as a crucial direction for future work.

Furthermore, the analysis would benefit from comprehensive ablation studies to better understand the sources of the performance gains. For instance, disentangling the separate impacts of weight and polarity guidance would clarify the mechanics of the learned heuristic.

A sensitivity analysis regarding GNN depth and RL hyperparameters would also be a valuable addition, providing a more complete picture of the model's robustness and helping to guide future work in this direction.

**Questions:**

1.	How stable is RLAF training across random seeds? Does GRPO converge consistently?

2.	Did the authors attempt fine-tuning across mixed problem distributions (e.g., 3SAT + 3COL)? Given the "solver-agnostic structural properties" observed, could the authors comment on whether transfer learning (pre-training a GNN policy on a general SAT solver/distribution and then fine-tuning for specific instances or solvers) is a promising avenue for reducing the training cost and improving generalization further?

---

> ### Author Response · Authors · 2025-11-21
> **Response to Reviewer CgoS**
>
> We thank the reviewer for the detailed and positive assessment of our work and for the constructive suggestions.
>
> Based on the feedback, we have expanded our ablations. In Section B.7 in the appendix, we have added an experiment that quantifies the individual contribution of weight and polarity guidance. Overall, the weight guidance accounts for most of the performance gain, while the effectiveness of polarity guidance is more data-dependent. In Section B.8, we have further added an analysis of how the GNN depth affects performance. Here, models with more layers yield more effective policies, suggesting that the policies exploit complex graph patterns of non-trivial depth.
>
> Let us further address the raised questions:
>
> **Q1:** We do observe consistent convergence of our GRPO-based training across different random seeds. To quantify this, we trained two additional policies to guide Glucose on 3SAT problems with different random seeds. Together with the original model from our main experiments, the mean number of decisions on 3SAT(300) is $307385.45 \\pm 2337.98$ across the three models. The relative deviation across these models is therefore less than one percent.
>
> **Q2:** We did attempt finetuning on mixed distributions, but did not observe positive transfer between different instance classes. Instead, we observed worse performance of the mixed policy across distributions compared to specialized guidance policies. The capacity of our moderately sized GNN may be insufficient to learn effective guidance strategies across multiple distributions with a single model. We did not try training models with mixed base solvers, but based on the observed correlations of learned variable weights, this may be a viable option for learning shared GNN policies for multiple solvers. Exploring the exact conditions that enable positive transfer learning across mixed instance classes and solvers remains future work.
>
> An orthogonal future research direction that could reduce training costs is self-supervised pretraining of the GNN \[1\]. This would teach the model to extract general structural features from graphs without requiring online calls to a SAT solver. Starting RL-finetuning from such a pretrained model may reduce the number of GRPO iterations required and may make training less sensitive to the specific choice of training problems.
>
> \[1\] Thakoor, Shantanu, et al. "Large-Scale Representation Learning on Graphs via Bootstrapping." *International Conference on Learning Representations, 2022*.

---

### Meta-Review · Area_Chair_SHMJ · 2025-12-20

**Summary:**

The paper describes a new technique to train a graph neural network by reinforcement learning to guide branching heuristics in satisfiability problems.  The main concerns of the reviewers are:

1. The limited scalability of the approach that was only demonstrated with small SAT problems
2. The limited ablation study
3. Lack of comparison with alternative distributions

**Reviewer Concerns:**

The last two concerns above have been addressed.  The authors expanded in the ablation study and included a sensitivity analysis with a comparison to alternative distributions.  The first concern about the limited scalability of the approach remains, but the reviewers did not view this as a deal breaker.  The performance gains on small instances is valuable in itself.  As for large industrial problems, a distributed multi-GPU setup would be needed, but such a setup is not available to everyone.

**Reviewer Scores:**

The reviewers unanimously recommended acceptance of the paper before the rebuttal.  After the rebuttal, I expect the reviewers to maintain or increase their scores since two of the three main concerns have been addressed.

Reviewer CgoS: The authors included expanded the ablation study as requested by the reviewer.  Hence, I expect the reviewer to maintain or increase its score.

Reviewer KMQK: I expect this reviewer to maintain its score since the main concern about the small instances was not addressed, but this concern was not a deal breaker.

Reviewer gUdh: I expect this reviewer to maintain its score since the main concern about the small instances was not addressed, but this concern was not a deal breaker.

Reviewer j7Rf: The authors included a sensitivity analysis that addressed the concern regarding the lack of comparison with alrternative distributions.  Hence, I expect the reviewer to maintain or increase its score.

Overall, the proposed approach is a strong contribution to the advancement of SAT solvers.  It is practical for small iinstances and it reduces the number of decision and wall clock time by a good margin.

---

### Decision · Program_Chairs · 2026-01-26

Accept (Poster)